# Collective dynamics of strain-coupled nanomechanical pillar resonators

J. Doster [1], S. Hoenl[1,3], H. Lorenz[2], P. Paulitschke[2] & E.M. Weig[1]*

Semiconductur nano- and micropillars represent a promising platform for hybrid nanodevices. Their ability to couple to a broad variety of nanomechanical, acoustic, charge, spin, excitonic, polaritonic, or electromagnetic excitations is utilized in fields as diverse as force sensing or optoelectronics. In order to fully exploit the potential of these versatile systems e.g. for metamaterials, synchronization or topologically protected devices an intrinsic coupling mechanism between individual pillars needs to be established. This can be accomplished by taking advantage of the strain field induced by the flexural modes of the pillars. Here, we demonstrate strain-induced, strong coupling between two adjacent nanomechanical pillar resonators. Both mode hybridization and the formation of an avoided level crossing in the response of the nanopillar pair are experimentally observed. The described coupling mechanism is readily scalable, enabling hybrid nanomechanical resonator networks for the investigation of a broad range of collective dynamical phenomena.

[1] Department of Physics, University of Konstanz, Universitätsstrasse 10, 78457 Konstanz, Germany. [2] Fakultät für Physik and Center for NanoScience (CeNS), Ludwig-Maximilians-Universität, Geschwister-Scholl-Platz 1, 80539 München, Germany. [3]Present address: IBM Research - Zurich, Säumerstrasse 4, CH-8803 Rüschlikon, Switzerland. *email: eva.weig@uni-konstanz.de

Nanomechanical pillar resonators[1–5] represent particularly versatile types of nanomechanical resonators. They allow for the convenient integration of multiple functionalities using semiconductor heterostructures[3,6,7], and for the dense integration into large arrays[8,9]. In recent years, the dynamics of both bottom-up and top-down fabricated nanomechanical pillar resonators has been explored[2,4,5,10]. Each pillar exhibits two orthogonally polarized fundamental flexural modes of similar eigenfrequency, which are typically not completely degenerate, even for the case of a circular cross section, as a result of fabrication imperfections[4]. Most experimental investigations focus on a single mechanical mode[2,10] or on the interplay of a single mechanical mode with another degree of freedom, e.g. an embedded quantum dot[3,7], an NV center spin[8], a surface acoustic wave[11], or an optical cavity mode[12]. Some scanning probe applications target the mechanical coupling of the two fundamental flexural modes of a single nanopillar induced by an external force field for vectorial force field sensing[4,5]. To date the intrinsic coupling of the flexural modes, or the coupling between the flexural modes of adjacent nanopillars has not been addressed.

At the same time, the quest of engineering a controlled intrinsic coupling between nanomechanical resonators is receiving an increasing amount of attention. At present, intrinsic coupling of adjacent beam or string resonators has been reported and relies on the strain distribution in a shared clamping point[13–15]. In a related approach, the membranes constituting the building blocks of nanomechanical phonon waveguides exchange energy through physical connections[16,17]. Note that the appearance of collective dynamical effects in a resonator array is enabled by a sufficiently large coupling strength, mitigating the unavoidable, and fabricational disorder-induced detuning of its elements[18].

Here, we translate the concept of strain-induced intrinsic coupling to vertically oriented nanomechanical pillar resonators sharing the same substrate[3–6]. We consider a pair of inverted conical nanopillar resonators like the one displayed in Fig. 1. The pillars are etched into a (100) GaAs substrate using reactive ion etching[2] and feature eigenfrequencies in the range of a few megahertz. In the following, we employ the strong coupling condition as a well-defined and experimentally easy-to-assess smoking gun demonstration of large inter-pillar coupling. Strong coupling, i.e. a coupling rate exceeding the linewidth of the resonances, is not required for most applications targeting the collective dynamics of nanopillar resonators. However, for realistic pillar parameters, fulfilling this condition is indicative of a sufficiently large coupling rate to beat the disorder in a nanopillar array.

## Results

**Mode hybridization.** We first investigate a nanopillar pair with bottom radius $r \approx 310$ nm, height $H \approx 7$ μm, and center-to-center distance $d \approx 1.3$ μm (see inset of Fig. 1). The taper angle of 1.1° is the same for all nanopillars discussed in this work. The nanopillar pair is driven at frequency $f_{drive}$ using a piezo actuator in a scanning electron microscope, allowing to image the resulting envelope of its mechanical vibration. In total, we find four well separated vibrational modes with no spectral overlap as shown in Supplementary Fig. 1, which are identified with the four eigenmodes of the pillar pair. With respect to the indistinguishable ⟨100⟩ crystal directions, these modes are in the following referred to as 'horizontal' (H) and 'vertical' (V) polarization of the left (L) and right (R) pillar, and correspondingly labeled LH, LV, RH, and RV (see Supplementary Fig. 1). One of these eigenmodes (LV) is shown in Fig. 2a, b, which display the vibrational envelope of the nanopillar pair driven at $f_{drive} = f_{LV}$, imaged from the top and in a tilted view. A more careful inspection of the mode reveals that in Fig. 2a, b not only the left resonator vibrates with a large amplitude but also the right resonator exhibits a simultaneous vibration, albeit with a much smaller amplitude (see also Supplementary Movies 1 and 2). This is also apparent from Fig. 2c which displays the vibrational amplitude of both pillars extracted from micrographs obtained at different drive frequencies $f_{drive}$ in the vicinity of the resonance $f_{LV}$. Clearly, the two amplitudes evolve simultaneously with the drive frequency $f_{drive}$, which implies the existence of a hybridized mode. The underlying coupling between the two pillars is mediated by the joint substrate.

**Avoided level crossing.** To obtain a more thorough understanding of the observed inter-pillar coupling, we measure the response of a second pillar pair with $r \approx 310$ nm, $H \approx 7$ μm, and $d \approx 1$ μm using the optical detection setup described in the Methods section. Thermal tuning, readily implemented by the laser used for optical detection, is employed to sweep the

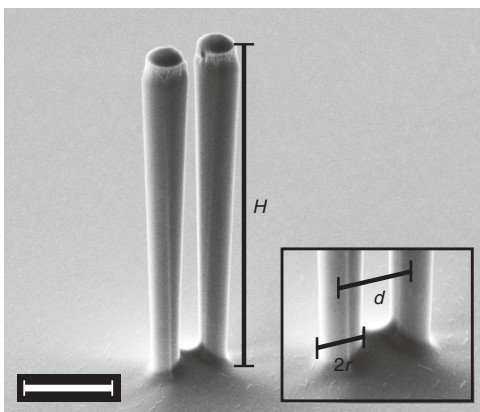

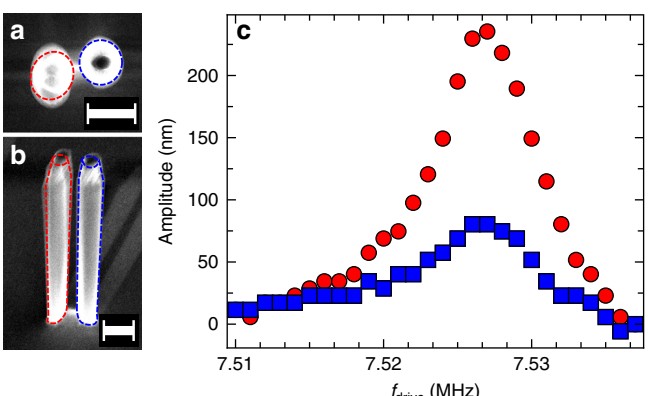

**Fig. 2** Hybridized mode. **a** Scanning electron micrograph of a resonantly driven ($f_{LV} = 7.527$ MHz) pair of nanopillars ($H \approx 7$ μm, $r \approx 310$ nm, $d \approx 1.3$ μm) imaged from the top and **b**, in a 60° tilted view. Red and blue dotted lines indicate the circumference of the undriven left and right resonator, respectively. **c** Amplitude for different drive frequencies $f_{drive} \approx f_{LV}$ of left (red circles) and right (blue squares) resonator determined from the scanning electron micrographs. Scale bar in **a** and **b** corresponds to 1 μm

**Fig. 1** Nanomechanical system. Scanning electron micrograph of a pair of nanopillar resonators with a height of $H \approx 7$ μm, a foot radius of $r \approx 310$ nm and a center-to-center distance of $d \approx 1.1$ μm in a 60° tilted view. The taper angle of 1.1° gives rise to a somewhat larger head radius. Scale bar corresponds to 2 μm. Inset shows a zoom of the clamping point of the two nanopillars

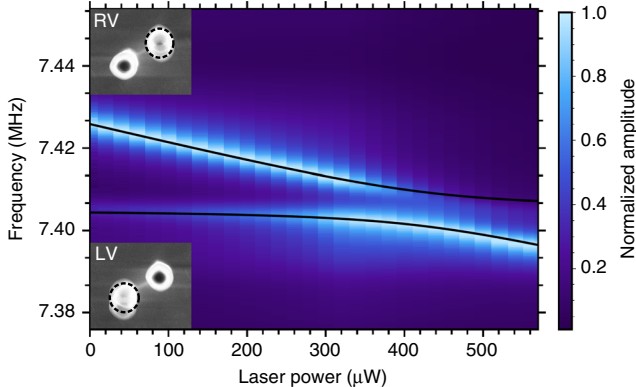

**Fig. 3** Strong coupling. Frequency response measurements of the right nanopillar under thermal tuning reveals an avoided crossing as an evidence for strong coupling. The black line shows a fit to the data, yielding a level splitting of $g/2\pi = 8.3$ kHz $> \Delta f \approx 3.5$ kHz, with the linewidth $\Delta f$. Insets show respective scanning electron micrographs of the two modes, indicating a vertical oscillation direction when the resonators are resonantly driven near $f_{LV}$ (lower inset) and $f_{RV}$ (upper inset), far from the avoided crossing

eigenfrequency of the higher-frequency pillar through that of the other pillar which remains largely unaffected (see Supplementary Note 3 for details). Figure 3 shows an avoided crossing of a pair of nanopillar resonators, which is indicative of strong mechanical coupling. A fit of the data using the model of two linearly coupled harmonic oscillators is also included as a black solid line[19]. It yields a coupling strength $g/2\pi = 8.3(18)$ kHz which exceeds the linewidth of the mechanical resonances $\Delta f \approx 3.5$ kHz. Hence, we demonstrate strong intrinsic coupling between the two nanopillar resonators, for this specific set of geometry parameters. The two modes to the left of the avoided crossing are assigned to the vertical vibration of each resonator (LV, RV) via scanning electron micrographs shown in the insets of Fig. 3, where the tuned pillar with the higher frequency corresponds to the right pillar. Further evidence for the inter-pillar coupling arises from the evolution of the vibration amplitudes in Fig. 3. Since the laser is focused on the frequency-tuned right pillar, mainly the vibration of this one resonator is detected and the vibration of the left pillar is only weakly resolved through the stray field of the laser. Clearly, the transition of the strong signal of the right pillar from the upper to the lower branch of the avoided crossing reflects the hybridization of the two pillars near their resonance at 425 μW and $f_r = 7.401$ MHz.

The observed strong coupling supports the conclusions drawn from Fig. 2, the data for which was acquired without frequency tuning the right pillar and with variable $f_{drive}$. An avoided crossing measured for this pair with $d = 1.32$ μm is included in Supplementary Fig. 1. The untuned situation probed in the scanning electron microscope measurements corresponds to the state of the system towards the leftmost edge of the avoided crossing: far from resonance, the individual eigenmodes of the two pillars dominate the response, however, a slight hybridization is already apparent as a consequence of their strong coupling.

**Geometry dependence of the coupling strength**. Strain-mediated coupling through the substrate is expected to depend on the geometry of the pillar pair. In particular, the bottom radius $r$ of the nanopillars, their height $H$ and their center-to-center distance $d$ have proven influential. In the following, we investigate the dependence of the nanopillar coupling strength on these parameters in measurements and finite element simulations, using the evaluation procedure detailed above.

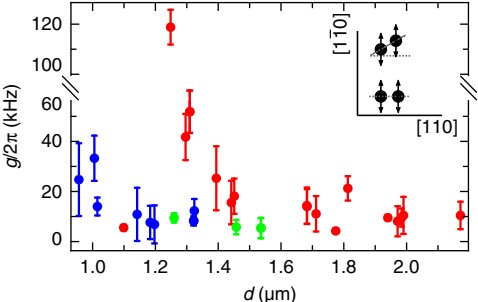

**Fig. 4** Measured geometry dependence of coupling strength. Experimentally determined coupling rate $g/2\pi$ of the vertical modes of the two nanopillars over their center-to-center distance $d$ for several samples and pillar pair geometries with $r \approx 430$ nm & $H \approx 7$ μm (red), $r \approx 335$ nm & $H \approx 7$ μm (blue), $r \approx 330$ nm & $H \approx 8.2$ μm (green). Error margins indicate the tolerance of the fitted coupling rate. The inset shows the vibration direction for differently oriented pillar pairs (see Supplementary Note 5 for details)

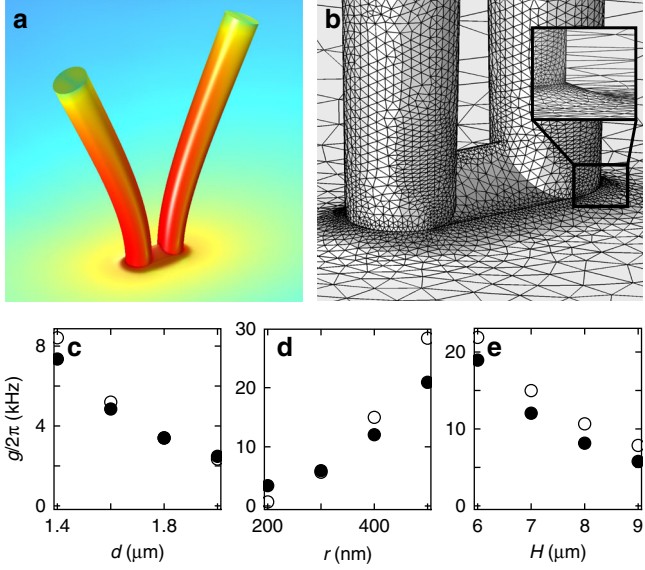

**Fig. 5** Finite element simulation of nanopillar pairs. **a** Simulation of the stress distribution between two (identical) nanopillar resonators vibrating in the antisymmetric hybrid mode. Red (blue) corresponds to high (low) stress, respectively. **b** Finite element simulation model, highlighting the realistic clamping area of the pillar pair as well as the narrowing mesh implemented near the pillar feet. The inset shows a zoom of the transition from a nanopillar to the substrate. **c**–**e** Finite element simulation of coupling strength of vertical (filled circles) and horizontal (empty circles) modes as a function of **c** center-to-center distance $d$ ($r = 400$ nm, $H = 7$ μm), **d** nanopillar bottom radius $r$ ($d = 2r + 400$ μm, $H = 7$ μm) and **e** nanopillar height $H$ ($r = 400$ nm, $d = 1.2$ μm). The taper angle for all simulations is 1°. We assume an isotropic substrate with Young's Modulus $E_{[100]} = 85.9$ GPa

Figure 4 shows the measured coupling strength for pillar pairs of different foot radius and height, plotted as a function of their separation. Clearly, the coupling strength increases with decreasing center-to-center distance, while the largest coupling rates are found for nanopillars with a larger foot radius. In order to validate these findings, finite element simulations are performed. An example for the case of identical pillars with realistic clamping conditions is shown in Fig. 5a, b. A significant overlap of the

stress distribution is observed, indicating the intrinsic, strain-mediated coupling between the pillars which is also reflected in the appearance of an antisymmetric hybrid mode.

Frequency tuning is incorporated by a variation of Young's Modulus $E$ of one of the nanopillars, mimicking thermal tuning. The level splitting $g/2\pi$ is then also obtained by fitting the model of the coupled oscillators to the simulation data (see Supplementary Note 4). We investigate the dependence of $g/2\pi$ on each of the parameters $d$, $r$, and $H$ while the other two parameters remain fixed. Only when sweeping the bottom radius $r$, $d$ is adapted accordingly to ensure a constant edge-to-edge distance.

Figures 5c–e display the simulated coupling strength as a function of the three parameters under investigation. Figure 5c depicts an increasing coupling when the center-to-center distance of the two nanopillars is decreased. This is a typical signature of strain coupling[20], and in agreement with the measurements in Fig. 4. Furthermore, Fig. 5d predicts an increasing coupling strength with the bottom radius of the two nanopillars, as a consequence of the strain caused at the pillar foot upon its deflection, and the experimental data in Fig. 4 clearly reflects this behavior. Finally, Fig. 5e displays a decrease of the coupling strength with the height of the pillars, again in consequence of the resulting strain profile. The dependence on the height, however, can not clearly be established in Fig. 4 since the differences in coupling for the two investigated pillar heights are smaller than the error bars.

The results shown Fig. 4 have been obtained for pillar pairs with different orientations on the substrate as indicated in the inset of Fig. 4 (see Supplementary Note 5 for details). It is expected that the angular dependence of Young's modulus of the crystalline GaAs substrate should lead to an angular dependence of the strength of the strain-induced coupling (see Supplementary Note 5). However, differences in coupling strength with pillar orientation are not experimentally resolved, likely because of the fabrication-induced disorder in the pillar geometry governing the uncertainty in our measurements. To validate the influence of the substrate's Young's Modulus, further studies should address the vibration polarization of hybridized (symmetric or antisymmetric) modes for arbitrary orientation of the pillar pair.

## Discussion

In conclusion, we establish a new nanomechanical platform for the investigation and direct visualization of collective behavior of coupled nanopillars. We reveal strain-induced coupling between two adjacent nanomechanical pillar resonators with coupling rates of up to 120 kHz, and show mode hybridization as well as the formation of an avoided level crossing under thermal tuning of one of the nanopillars. The coupling strength is found to depend on the center-to-center distance of the nanopillars, as well as their diameter and height. Numerical finite element simulations reproduce the observed scaling, confirming that the coupling between two nanopillar resonators is mediated by strain in the substrate which acts as a shared clamping point.

The established inter-pillar coupling can be readily employed for hybrid nanopillars e.g. based on III–V semiconductor heterostructures[3,6,7,21,22] featuring quantum dots, quantum wells, or integrated cavities. This may open the path towards the direct, mechanically mediated synchronization of single-photon sources, e.g. for applications in quantum information processing[23]. In addition, the scheme is scalable to a large number of pillars in a straightforward way, since vertically oriented nanopillars are ideally suited for dense integration, while a large coupling rate mitigates their unavoidable disorder-induced detuning. Assuming a typical quality factor of 2000 and a realistic disorder of ~1%, this condition is fulfilled for a coupling rate exceeding the linewidth by about an order of magnitude, a ratio which has been accomplished for several of the investigated pillar pairs. In addition we envisage to implement in-situ thermal tuning of the pillar eigenfrequencies in an array using a spatial light modulator[24]. The resulting nanomechanical network offers inherent, acoustically mediated nearest-neighbor coupling[25]. The large vibration amplitudes of singly-clamped nanopillars in the range of tens or hundreds of nanometers allow for the microscopic imaging of their response and thus for the direct visualization of the many-body dynamics in an all-mechanical array. This promises important insights for the emergent field of collective dynamical phenomena which includes phenomena such as acoustic metamaterials[17,26,27], synchronization[28–31], topologically protected transport[27,32,33], or non-reciprocal signal transduction[34], and may pave the way towards nanomechanical computing[35] or nanomechanical implementations of neural networks [36].

## Methods

**Fabrication**. The circular nanopillar heads are defined by electron-beam lithography with 5 kV acceleration voltage and 30 μm aperture on a GaAs substrate with spin-coated polymethyl methacrylate resist. The pattern transfer is performed via electron-beam evaporation of nickel and subsequent lift-off. A final anisotropic inductively coupled plasma reactive ion etching (ICP-RIE) process reveals the inverted conical nanopillars. Anisotropic etching is achieved with a gas flow of 13-sccm silicon tetrachloride and 0.2-sccm nitrogen, a RF power of 65 W and an ICP power of 50 W.

**Measurement details**. Measurements are performed in vacuum $<10^{-4}$ mbar and at room temperature using piezo-actuation, while the response of the nanopillars is read-out by scanning electron microscope imaging[37] or by optical detection[3], focusing a laser with $\lambda = 635$ nm wavelength onto the head of one nanopillar and detecting the vibration-induced modulation of its reflection.

## Data availability

Data and analysis code are available at Zenodo [https://doi.org/10.5281/zenodo.3522940].

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

## Acknowledgements

Financial support from the European Unions Horizon 2020 programme for Research and Innovation under grant agreement No. 732894 (FET Proactive HOT), the Deutsche Forschungsgemeinschaft via the Collaborative Research Center SFB 767, the Volkswagen Foundation through grant Az I/85 099, as well as the German Federal Ministry of Education and Research through contract no. 13N14777 funded within the European QuantERA cofund project QuaSeRT and the programm "Validation of the Technological Innovation Potential of Scientific Research - VIP" is gratefully acknowledged.

## Author contributions

Measurements were performed by J.D., while sample fabrication was done by J.D., S.H., and P.P.; measurement results and data interpretation were discussed by all authors.

## Competing interests

The authors declare no competing interests.
