## [Peer Review File · Nature Communications]

Reviewers' Comments:

Reviewer #1:

Remarks to the Author:

The manuscript by Doster et al. presents experimental results that evidence coupling between closely spaced nanomechanical pillar resonators. The experiments are carefully executed and well described. The nanopillar platform is certainly interesting for various applications, such that these studies are warranted. I thus support their publication in Nature Communications, if the authors can clarify the points below:

- Most importantly, I would like to ask if the authors can describe more explicitly why the realization of strong mechanical coupling in this particular platform is needed: What specific applications or follow-up research on nanopillars are enabled by these results? Of course, coupling of mechanical vibrations has been extensively studied in mechanical systems of all sizes, including nanomechanical resonators. As such, its observation in itself is not necessarily surprising. To understand the value of this work, readers would benefit from explicitly hearing what the most promising opportunities of coupled nanopillars are.

- Related, could the authors provide more context for the obtained magnitude of the coupling rate? I imagine that some applications benefit from coupling rates that exceed frequency disorder rather than linewidth, etc.

- What is the role of the exact clamping point geometry in the magnitude of the coupling rate? The main text does not comment on this, and I can imagine that the 'ridge' between the two pillars plays a significant role. In this supplementary, the authors state that it is important to take this shape into account in simulation, which makes me wonder what its impact on the experimental results is.

Reviewer #2:

Remarks to the Author:

Doster et al. report in their manuscript on experimental work that demonstrates coupling of two nanomechanical pillar resonators mediated by strain. The authors report on fabrication of nanomechanical pillar resonators in GaAs using reactive ion etching. They engineer two neighboring pillars with height in the sub-10 μm regime of about 600nm diameter and about 1 μm separation that are vibrating at MHz frequencies. They infer the resonance frequency of these nanomechanical resonators by excitation using a piezoelectric transducer and subsequent inspection of the mechanical displacement by an SEM (oscillation envelope) or an optical measurement using a laser. The two lowest eigenfrequencies of each nanopillar are closely inspected and attributed to the vertical and horizontal vibration direction. These two modes can couple to the same two modes in a neighboring pillar once the oscillation energies are made similar, i.e., by entering the strong coupling regime. The authors demonstrate this regime by tuning the resonance frequency of one of the pillars using laser-induced heating that decreases the mechanical frequency. In this way an avoided crossing in resonance frequencies is demonstrated. The authors extract a coupling of $\sim 8\text{kHz}$ that is larger than the mechanical dissipation rate of $\sim 3.5\text{kHz}$. The coupling between the two pillars is traced back to the common substrate that mediates a strain-induced coupling mechanism. The authors model this coupling using FEM methods. They show that they can reproduce the predicted increase of coupling strength with increasing radius of the pillars or with decreasing distance in the experiment. However, the simulated decreased coupling strength with increasing height can not be supported by the data.

Overall, this manuscript presents novel and original work in the field of nanomechanics. The work demonstrates strong coupling of two nanomechanical resonators mediated by a common support that results in strain-mediated coupling. The authors demonstrate the strong coupling regime by tuning the resonance frequency of one of the resonators by laser-induced heating. The presented data is scientifically sound and supported by detailed FEM simulations. The manuscript is written in a clear and comprehensible way and relevant and interesting to the specialist working in the field.

Although the work is novel and scientifically interesting, I have a criticism towards the impact of the presented work and the scalability of the presented approach. Hence, I would not recommend publication of the manuscript in Nature Communications in the present state.

Strong coupling in nanomechanical resonators has been demonstrated before, see, e.g., the cited Refs. [23,24]. I therefore feel that the authors should motivate, why their presented approach is superior to the ones recently published.

Further, the authors rely on the scalability argument for making the work attractive to the field, i.e., the abstract mentions: "...readily scalable, enabling hybrid nanomechanical resonator networks for the investigation of a broad range of collective dynamical phenomena." However, scalability in this system of nanopillar resonators seems to be practically infeasible. How would this nanopillar system allow for individual tunability of each resonator in a large network? I would highly doubt that tuning each nanomechanical resonator individually by laser-induced heating would lead to a scalable approach of an interconnected nanomechanical network of resonators. The authors should clarify how they envision realization of individual and practically feasible in-situ frequency tuning of a large network of resonators.

Furthermore, the coupling strength between the nanomechanical resonators is only by a factor of about 3 larger than the mechanical dissipation rate. What is the origin of the strength of dissipation and how can the dissipation be engineered?

Finally, a minor question concerns the explanation of the meaning of the shaded region in Fig. 4? What do the authors indicate with that region?

Reviewer #1

We thank the reviewer for their positive assessment of our work, and for their constructive comments. In the following we'd like to discuss the points which demanded clarification, along with the corresponding changes we've made to the manuscript.

1. *Most importantly, I would like to ask if the authors can describe more explicitly why the realization of strong mechanical coupling in this particular platform is needed[...]*

Indeed, the majority of future applications of the nanopillar platform do not necessarily require coupling rates exceeding the mechanical linewidth, i.e. strong coupling¹. In general, the observation of collective dynamics at large rather demands coupling rates larger than the frequency disorder (*see also 2.*).

Unavoidable fabrication-induced deviations of the ideal pillar geometry lead to a frequency disorder at the level of 1% of the resonance frequency. For nanopillars with a quality factor of order 1,000 the strong coupling condition imposes a bound which is getting close to the required threshold. We therefore chose to employ the strong coupling condition as a well-defined and experimentally easy-to-assess smoking gun demonstration of nanopillar coupling strong enough to beat the disorder in an array. (Note that for the case of a nanopillar pair, geometric effects (*see also 2.*) lead to systematic shifts of the pillar eigenfrequencies exceeding the net frequency disorder. Therefore, a direct comparison of the coupling strength and the observed eigenfrequency differences is not meaningful for our experiment.)

Following the reviewer's feedback we have realized that these aspects have not been made clear in the initial version of the manuscript. We have modified the introduction of the revised manuscript (p. 3-4) in order to fully take them into account. We have also added the above estimation (p. 11). In addition, we propose to avoid the notion of strong coupling in the title of the manuscript by modifying it to "Collective dynamics in strain-coupled nanomechanical pillar resonators" (p. 1).

2. *Related, could the authors provide more context for the obtained magnitude of the coupling rate? I imagine that some applications benefit from coupling rates that exceed frequency disorder rather than linewidth, etc.*

See above. Considering frequency disorder, there are several points to be raised: As already remarked, the frequency spread in a nanopillar pair cannot directly be related to the frequency disorder in an array. This is a consequence of our diffusion-limited reactive ion etching process, which yields larger etch rates for more 'exposed' pillars (e.g. at the edge of an array) and smaller etch rates for more 'shielded pillars' (e.g. in the center of an array). For the case of a pillar pair, the broken symmetry of each pillar's surroundings gives rise to the connecting ledge between the pillars (*cf. inset of Fig. 1 of main text*). This in turn produces a large splitting between the horizontal (H) and vertical (V) modes of the pair, which is clearly reflected in the experimental results as well as in numerical simulations.

The characterization of the frequency disorder in pillar arrays goes beyond the scope of this work. Preliminary data suggests a frequency disorder of 1.5% in the center of a pillar array,

¹ Only a few examples of strongly coupled resonator arrays are discussed in the literature in the context of subradiant behavior, see e.g. [1,2].

where the etching rate can be considered homogeneous. In this case the disorder is predominantly caused by imperfections in the electron-beam lithographic definition of the pillar heads. After some careful process optimizations, we have been able to suppress deviations from the circular shape of the pillar heads to below 1%. The new samples remain to be characterized, but we expect a corresponding reduction of the frequency disorder.

The added estimation (p. 11) is based on these parameters.

3. *What is the role of the exact clamping point geometry in the magnitude of the coupling rate? [...]*

In order to quantify the effect of the pillar pair's clamping area on the coupling rate, we have performed numerical simulations using Comsol Multiphysics. The attached figure shows a comparison of the coupling rate between two nanopillars on a flat substrate (left) and two nanopillars with a realistic clamping area including the 'ridge' (right). The results demonstrate that the general magnitude of the coupling rate doesn't change significantly with or without the ridge, but equalizes the in-plane and out-of-plane coupling rate. Without the ridge the coupling rate of the horizontal modes consistently exceeds that of the vertical modes, which is in line with the phenomenological picture of an increased strain overlap in the former configuration. The ridge seems to alter the strain distribution, resulting in almost identical coupling rates for H and V modes.

Figure: Numerically simulated coupling rate of the horizontal (H) and vertical (V) modes as a function of the pairs' center-to-center distance. The simulated pillar pair with $r=400$ nm, $H=7$ μm and taper angle 1° is simulated for two different clamping conditions. (a) Nanopillar pair on a flat substrate. (b) Nanopillar pair with realistic clamping conditions. Hollow (filled) circles correspond to H (V) modes, respectively.

We have included this figure along with a brief discussion of the effect of the clamping geometry on the coupling rate on page 9 of the revised Supplemental Information.

Unfortunately, the error margins of the experimentally determined coupling rates (*see e.g. Fig. 4 of main text*) are too large to validate these numerical findings.

References:

- [1] Jenkins S. D., Ruostekoski J., Papasimakis N., Savo S. & Zheludev N. I. Many-Body Subradiant Excitations in Metamaterial Arrays: Experiment and Theory. *Physical Review Letters* **119**, 053901 (2017). URL <https://doi.org/10.1103/physrevlett.119.053901>.
- [2] Watson D. W., Jenkins S. D., Fedotov V. A. & Ruostekoski J. Point-dipole approximation for small systems of strongly coupled radiating nanorods. *Scientific Reports* **5**, 5707 (2019). URL: <https://doi.org/10.5258/SOTON/D0856>.

Reviewer #2

We thank the reviewer for their thorough assessment of our work, and appreciate their judgement of our work as being “*novel and original*”, “*scientifically sound*”, “*novel and scientifically interesting*”, as well as “*written in a clear and comprehensible way*”. However, the reviewer raises criticism towards the impact and the scalability of the presented work. In the following, we would like to take the opportunity to address those concerns. We hope that the following discussion and the revised version of the manuscript will help dispel the reviewer’s doubts, and kindly ask for a reconsideration of our work for publication in Nature Communications.

1. *Strong coupling in nanomechanical resonators has been demonstrated before, see, e.g., the cited Refs. [23,24]. I therefore feel that the authors should motivate, why their presented approach is superior to the ones recently published.*

We agree with the reviewer that strong coupling between nanomechanical resonators has been accomplished in [23,24]. The platform under investigation in these studies is based on doubly-clamped silicon nitride string resonators which are connected via specifically engineered clamping regions. In our opinion the justification for the presented platform of vertically oriented GaAs nanopillars is threefold:

(1) GaAs nanopillars are a unique platform for hybrid devices. They lend themselves to the integration of semiconductor heterostructures, enabling coupling to e.g. excitonic or polaritonic excitations in quantum dots, or to electromagnetic or acoustic cavity modes between distributed Bragg reflectors, just to name a few. While the potential of semiconductor nanopillars is appreciated in a vast research community, the current state of the art is restricted to single pillar implementations. The coupling of pillars which can be employed to enhance the individual building block’s performance or to enter the novel field of hybrid metamaterials has not been taken into account yet.

(2) The vibrational modes of nanopillar resonators can easily be driven to amplitudes of tens and hundreds of nanometers which is easily resolved with a microscope. This opens up a radically simple and straightforward in-situ readout technique for the pillars even in large arrays. Even more, it allows to directly visualize the two-dimensional collective response of such arrays which is extremely challenging in other microscopic implementations.

Figure: Direct visualization of the response of a 1D-coupled nanopillar array with an optical microscope.

This represents an asset which is not contained in any other micro- or nanoresonator array. This is also true for silicon nitride nanostrings. Their amplitudes range in the sub-nanometer regime, and rarely exceed a few nanometers for extremely nonlinear driving, rendering direct visualization impossible. For nanostring resonators, techniques such as optical interferometry or electrical displacement detection techniques are employed. Both are, in principle, scalable to resonator arrays, albeit under severe complications.

(3) Nanopillar resonators can easily be scaled to large arrays as a result of their vertical orientation as well as single-lithography step processing. See below for more details.

All three aspects have been discussed in the initial version of the manuscript. In order to avoid over-emphasizing the role of strong coupling, we have removed this notion from the title of our manuscript (p. 1). The *superior* properties of coupled nanopillars for future investigations have been emphasized in the conclusion (p. 11-12) of the revised manuscript.

2. *Further, the authors rely on the scalability argument for making the work attractive to the field [...] However, scalability in this system of nanopillars resonators seems to be practically infeasible. How would this nanopillars system allow for individual tunability of each resonator in a large network? I would highly doubt that tuning each nanomechanical resonator individually by laser-induced heating would lead to a scalable approach of an interconnected nanomechanical network of resonators. The authors should clarify how they envision realization of individual and practically feasible in-situ frequency tuning of a large network of resonators.*

We appreciate the reviewer's doubts on the scalability of the nanopillar platform based on the unavoidable frequency disorder. Indeed, as detailed in the response to Reviewer #1, we agree that fabrication-induced deviations from the ideal pillar geometry always lead to a non-negligible spread in eigenfrequencies. As also discussed in the context of the response to Reviewer #1, we emphasize that the distribution of eigenfrequencies in the nanopillar pair reported in the manuscript is not limited by the disorder, but by systematic shifts between the horizontal (H) and vertical (V) modes of the pillars arising from the diffusion-limited etching in combination with a non-isotropic pillar environment. Preliminary results suggest that the eigenfrequency disorder of actual nanopillar arrays can be reduced to approx. 1%.

To mitigate the remaining deviations in the nanopillars' eigenfrequencies, we propose the following two measures:

(1) Large coupling rates: The detrimental effect of the disorder can be compensated by large enough inter-pillar coupling rates. The condition to fulfil for most applications of resonator arrays is a coupling rate exceeding the disorder. It is this aspect which actually convinced us to submit our work to a Nature Communications: The demonstration of strong coupling between the two nanopillars, along with the developed understanding about the scaling of the coupling strength with geometric parameters, paves the way to the realization of nanopillar arrays in which this condition is fulfilled. As also pointed out in the response to Reviewer #1, it is not the mere observation of strong coupling which is, in our point of view, the most striking result. We rather use the strong coupling condition as a well-defined and experimentally easy-to-assess smoking gun demonstration of nanopillar coupling strong enough to envision beating the disorder in an array (given typical parameters discussed in response to Reviewer #1). In this case, a tuning of individual nanopillar eigenfrequencies would not be required.

(2) Eigenfrequency tuning: In addition, we agree with the notion of the reviewer that a nanopillar eigenfrequency tuning mechanism would be useful. However, unlike the reviewer, we are convinced that laser-induced frequency tuning remains a viable method even for a large array.

(2a) Coarse tuning can be accomplished by irreversible laser-induced tuning at powers exceeding the range reported in Fig. 3. Figure S2 of the Supplementary Information indicates that at very large laser power, irreversible eigenfrequency stiffening occurs.

(2b) Fine tuning can subsequently be accomplished by reversible eigenfrequency softening induced by heating as exploited in the context of this manuscript. In order to address all pillars of the array simultaneously, we propose to take advantage of a spatial light modulator (see e.g. [1]). These commercially available devices allow to distribute and control the laser power impinging on a pixel array, which can be designed to match the pitch of the pillar array.

Those considerations have been included on p. 11 of the revised manuscript, where the prospect

of scaling to larger arrays is discussed.

3. *Furthermore, the coupling strength between the nanomechanical resonators is only by a factor of about 3 larger than the mechanical dissipation rate. What is the origin of the strength of dissipation and how can dissipation be engineered?*

The observed room temperature quality factors of a few thousands lie well within the values reported in literature for crystalline, singly clamped resonators [2,3] and, more generally, for most nanoscale resonators which do not incorporate tensile stress (see e.g. Fig. 2 of review [4]). According to the very detailed study in [5] intrinsic material losses (two-level fluctuators both on the surface and in the bulk) are likely limiting the quality factor in GaAs resonators.

Note that this finding is in line with the results of [6,7] on the *intrinsic* quality factors of stressed silicon nitride devices (nanostings and membranes) which are of the same order of magnitude and equally limited by intrinsic (surface) loss. However, as a result of dissipation dilution, i.e. the strongly increased stored mechanical energy under tensile stress, the measured quality factors of silicon nitride strings amount to several 100,000 – but are entirely owed to the tensile stress.

To decrease the dissipation in GaAs nanoresonators, especially the operation at cryogenic temperatures has proven effective, and can, in principle, be incorporated with our experiments.

4. *Finally, a minor question concerns the explanation of the meaning of the shaded region in Fig. 4? What do the authors indicate with that region?*

The gray region in Fig. 4 was intended to serve as a guide to the eye for the over-all decay of the coupling rates with larger inter-pillar distances. Following the reviewer's comment, we have decided to omit the gray shaded region in Fig. 4.

References:

- [1] Sokolov, S. *et al.* Tuning out disorder-induced localization in nanophotonic cavity arrays. *Optics Express* **25**, 4598–4606 (2017). URL <https://doi.org/10.1364/OE.25.004598>.
- [2] Paulitschke, P., Seltner, N., Lebedev, A., Lorenz, H. & Weig, E. M. Size-independent Young's modulus of inverted conical GaAs nanowire resonators. *Applied Physics Letters* **103**, 261901 (2013). URL <https://doi.org/10.1063/1.4851897>.
- [3] Yeo, I. *et al.* Strain-mediated coupling in a quantum dot-mechanical oscillator hybrid system. *Nature Nanotechnology* **9**, 106–110 (2014). URL <https://doi.org/10.1038/nnano.2013.274>.
- [4] Imboden, M. & Mohanty, P. Dissipation in nanoelectromechanical systems. *Physics Reports* **534** 89–146 (2014). URL <https://doi.org/10.1016/j.physrep.2013.09.003>.
- [5] Hamoumi, M. *et al.* Microscopic Nanomechanical Dissipation in Gallium Arsenide Resonators. *Physical Review Letters* **120**, 223601 (2018). URL <https://doi.org/10.1103/PhysRevLett.120.223601>.
- [6] Verbridge, S. S., Parpia, J. M. High quality factor resonance at room temperature with nanostings under high tensile stress. *Journal of Applied Physics* **99**, 124304 (2006). URL <https://doi.org/10.1063/1.2204829>.
- [7] Villanueva, L. G. & Schmid S. Evidence of Surface Loss as Ubiquitous Limiting Damping Mechanism in SiN Micro- and Nanomechanical Resonators. *Physical Review Letters* **113**, 227201 (2014). URL <https://doi.org/10.1103/PhysRevLett.113.227201>.

Reviewers' Comments:

Reviewer #1:

Remarks to the Author:

I have read the response of the authors to my questions and reviewed the changes to the manuscript. In my opinion, these address all the comments raised and I thus support publication of this manuscript.

Reviewer #2:

Remarks to the Author:

The reply and revised manuscript of Doster et al. convincingly address the criticism I raised towards the scalability of their approach and the placement of their work in the context of strong coupling.

I can therefore recommend publication of the revised manuscript in Nature Communications. The work of Doster et al. represents a novel experiment in the field of collective nanomechanical behavior and opens up new possibilities in the field through coupling of a large array of nanopillar oscillators, which can be monitored all-optically in a straight-forward way and allows for coupling to embedded quantum systems, such as quantum dots.